# Do Bloggers Have Better Mental Health? The Social, Cognitive, and Psychological Benefits of Blogging in Emerging Adulthood

**DOI:** 10.3390/ijerph20085493

**Published:** 2023-04-13

**Authors:** Imge Tekniker, Rebecca Y. M. Cheung

**Affiliations:** 1School of Psychology and Clinical Language Sciences, University of Reading, Reading RG6 6ES, UK; 2Centre for Child and Family Science, The Education University of Hong Kong, Hong Kong SAR, China

**Keywords:** blogging, social support, memory slips, mental health

## Abstract

Background: The purpose of this study was to investigate the longitudinal processes by which blogging-related disclosure is linked to mental health. It was hypothesized that blogging had both social and cognitive benefits, including greater perceived social support and fewer memory slips, which were then associated with better mental health. Methods: A total of 194 emerging adults were recruited three times at approximately three months apart. Participants filled out a self-report about their blogging activities and perceived benefits, social support, memory, and mental health at each time point. Results: Path analysis indicated that perceived blogging-related benefits, needs, and traits mediated the relation between frequency of blogging and social support and memory slips, respectively. Moreover, social support marginally predicted greater mental health, whereas memory slips predicted poorer mental health, after controlling for baseline mental health, age, and gender. Conclusions: This study established the longitudinal associations between blogging and its benefits that may be vital for emerging adults’ mental health.

## 1. Introduction

Three decades of research has shown that written disclosure is linked to improvements in physical and mental health [1,2,3,4,5,6]. More recently, people have begun posting written self-disclosures on online platforms [7,8]. Indeed, digital communication platforms have expanded people’s opportunities to communicate their emotional experiences with a wider audience through blogging or blog writing [8].

### 1.1. Psychological Benefits of Blogging

As an avenue of self-disclosure, blogging gives people a chance to express their emotions, engage in self-talk, and organize their thoughts, all of which are associated with psychological and emotional health [9,10,11]. Additionally, blogging sometimes serves as a therapeutic aid by allowing people to give their life experiences meaning and coherence [10]. According to Miura and Yamashita [12], blogging is linked to better mental health through reflections and disclosure. The opportunity to reflect provides people with a better understanding of their situation and a better capability for reasoning their own actions [12], which are crucial for decision-making and revisions of their own attitudes. Through a deeper understanding and reflections, blogging can further foster feelings of satisfaction and meaning in life [12,13]. Extending these findings to the Chinese context, previous research indicated that blogging is common in Hong Kong [14], particularly among undergraduate students who are typically emerging adults [15]. Notably, in a sample of second-year undergraduate students, blogging also provided a useful avenue for bloggers in Hong Kong to engage in problem-solving, reflection, construction of knowledge, and communication of emotions [16].

### 1.2. The Role of Social Support

Social support may be a crucial process through which blogging is linked to better mental health. According to the model put forth by Miura and Yamashita [12], blogging provides people with social advantages including access to social support [10,12,17,18]. The self-disclosive nature of blogs, which enables one to communicate their own sentiments, concerns, and opinions with readers, allows bloggers to divulge their experiences without having to go through the discomfort of doing so in person [19]. Bloggers may also have the opportunity to receive the readers’ comments that serve as feedback and a mutual communication tool [10,17,18]. Through the interactivity of blogs, people can create an online community where they share support based on their personal experiences [19]. That is, the social support attained by blogging gives people access to an audience to interact with and further share their experiences [20]. For instance, according to Keating and Rains [21], bloggers who receive social support from their readers are better able to cope with the unpredictable nature of their illness by getting in touch with other people who experience a similar condition [21]. As another example, blogging is advantageous for people with chronic pain as they could have an easily accessible online social network [22]. In addition, in a sample of undergraduate students in Hong Kong, blogging served as a useful avenue to foster mutual support and communication [23]. By establishing a social support network, people often also experience better mental health [24].

### 1.3. The Role of Memory Slips 

The experience of fewer memory slips may be another process between blogging and enhanced mental health. According to Colman [25], memory slips are the minor errors made throughout the recollection process. These errors entail the experience of reality disorientation and difficulties in appropriately differentiating between reality and fantasy [25]. Previous research indicated that writing helped patients in an intensive care unit to fill in the memory gaps while they were in the hospital, thereby reducing their memory slips. Blogging also allows people to re-experience sensory events, which enables them to have a better understanding of the type of information being stored in their memory [26]. Blogs also serve as a physical memory aid for people, as it can be accessible from any location with an Internet connection. By recollecting specific experiences, blogging enhances memory and reduces memory slips. Having fewer memory slips may also be associated with better mental health. For instance, previous research indicated that memory problems worsen anxiety and depressive symptoms, as well as decrease self-assurance [27,28]. As memory performance diminishes, people experience worsened emotional health. As such, perceived memory complaints are often sensitive predictors of future depressive symptoms [29].

### 1.4. The Present Study

This longitudinal study aimed to investigate the perceived blogging-related benefits, needs, and traits, social support, and memory slips as mediators between blogging frequency and mental health in emerging adulthood, over and above age and gender as covariates. Indeed, computer-mediated communication platforms have become an avenue for virtual social support among emerging adults [30], who often experience frequent transitions and changes (e.g., residence, job, and education) and find themselves with little social support [31,32]. Blogs not only allow emerging adults to acquire social support, but also provide a space for reflections of their identity, instability, and possibilities [16,23]. In this study, we hypothesized that Time 1 (T1) blogging frequency would be positively related to T1 blogging-related benefits, needs, and traits. We also hypothesized that T1 blogging-related benefits, needs, and traits would be further related to Time 2 (T2) social support and memory slips. Finally, we hypothesized that T2 social support and memory slips would be related to Time 3 (T3) mental health, over and above the effects of T1 mental health, age, and gender. To examine the reversed directionality of effects between perceived blogging-related benefits, needs, and traits and blogging frequency, i.e., whether perceived blogging-related benefits, needs, and traits would predict blogging frequency and subsequent social support, memory slips, and mental health, supplementary analyses were further conducted.

## 2. Materials and Methods

### 2.1. Participants

A total of 194 emerging adults were recruited to participate in a larger study (masked for peer review) via online social platforms and mass emailing. Participants ranged from 18 to 27 years old (*M_age_* = 21.08, *SD_age_* = 2.01; 90.2% female). They were recruited to complete a packet of questionnaires for three times, with a three-month lag between time points. The rate of retention was 92.78% from Time 1 (T1) to Time 2 (T2), with *n_T2_* = 180, and 92.78% from T2 to Time 3 (T3), with *n_T3_* = 167. The present study was approved by the Human Research Ethics Committee at The Education University of Hong Kong. Informed consent was obtained before the beginning of this study. At the end of the study, participants received a maximum of HK$200 (~US$25.64) supermarket coupon as compensation for their time. Data were missing completely at random (MCAR), as indicated by Little’s MCAR test with χ^2^ (1671) = 1697.74, *p* = 0.32.

### 2.2. Measures

#### 2.2.1. Frequency of Blogging

At T1, a 1-item measure was created to examine participants’ blogging frequency on a 5-point scale from 0 (never) to 4 (almost every day). A greater score indicated a greater frequency of blogging.

#### 2.2.2. Blogging-related Benefits, Needs, and Traits

At T1, a 14-item measure of blogging [11] was used to assess the benefits of blogging, reassurance seeking, and private self-consciousness on a 5-point scale, ranging from 1 (strongly disagree) to 5 (strongly agree). The measure had four subscales including benefit to self, benefit to relationships with others, private self-consciousness, and reassurance seeking. Sample items included, “I can sort out my feelings through blogging” (benefits to self; totaling 3 items), “I can communicate with others honestly through blogging” (benefits to relationship with others; totaling 5 items), “I reflect about myself a lot” (private self-consciousness; totaling 3 items), and “It is important for me to receive positive comments from the people I feel close to” (reassurance seeking; totaling 3 items). Three additional items were included, namely, “I can express myself fully through blogging”, “Other people can fully understand me through my blog”, and “I would like to continue blogging”). The item scores were averaged, with higher averaged scores indicating greater perceived blogging-related benefits, needs, and traits. In this study, the measure had adequate internal consistency at α = 0.93. Confirmatory factor analysis of the 17 items indicated that the model fit acceptably to the data, χ^2^(109) = 243.69, *p* < 0.001, CFI = 0.91, TLI = 0.89, and SRMR = 0.07.

#### 2.2.3. Social Support

At T2, the 12-item Multi-dimensional Scale of Perceived Social Support (MSPSS) [33] was used to assess the perceived availability of social support from family, friends, and significant others on a 7-point scale, ranging from 1 (very strongly disagree) to 7 (very strongly agree). The MSPSS comprises three subscales (Family Support, Friend Support, Significant Other Support), each consisting of four items. Sample items included, “There is a special person who is around when I am in need” and “I have a special person who is a real source of comfort to me.” The item scores were averaged, with higher averaged scores indicating greater perceived overall social support. In this study, the measure had adequate internal consistency at α = 0.89.

#### 2.2.4. Memory Slips

At T2, the Prospective and Retrospective Memory Questionnaire (PRMQ) [34] was used to measure self-reported challenges in prospective and retrospective memory on a 5-point scale, ranging from 1 (never) to 5 (always). Sample items included, “Do you decide to do something in a few minutes’ time and then forget to do it?”, “Do you forget something that you were told a few minutes before?”, and “Do you fail to recall things that have happened to you in the last few days?” The item scores were averaged, with higher averaged scores indicating greater memory difficulties. In this study, the measure had adequate internal consistency at α = 0.93.

#### 2.2.5. Mental Health

At T1 and T3, the 14-item Mental Health Continuum Short Form (MHC-SF) [35] was used as a measure for well-being over the last month. A 6-point scale ranging from 1 (never) to 6 (every day) was used, with sample items including, “how often did you feel satisfied with life” (emotional well-being), “how often did you feel that that you had something important to contribute to society (social well-being) and “how often did you feel that your life has a sense of direction or meaning to it” (psychological well-being). The MHC-SF was validated in a Chinese adolescent sample and yielded good validity and reliability [36]. The measure had adequate internal consistency at T1 (α = 0.93) and T3 (α = 0.95).

#### 2.2.6. Data Analysis

Zero order correlations, means, and standard deviations were conducted using IBM SPSS Statistics version 29 for all variables. Path analysis was then conducted using MPLUS, Version 8.3 [37] to investigate the relations between T1 frequency of blogging, T1 blogging-related benefits, needs, and traits, T2 social support, T2 memory slips, and T3 mental health, over and above the effects of gender, age, and T1 mental health on T3 mental health. Maximum likelihood method was employed to examine the fit of the model to the observed matrices of variance and covariance. Full information maximum likelihood estimation was employed to handle missing data. Mediation effects were further examined using bootstrapping, given that bootstrapping yields more accurate estimates of the indirect effect standard errors than do other methods [38].

## 3. Results

Table 1 shows the correlations, means, and standard deviations for all variables. Among the participants, 34.02% reported that they did not blog and therefore did not complete the 14-item measure of blogging [11]. In addition, 17.01% reported that they blogged a few times every 3 months, 13.92% reported that they blogged a few times a month, 13.40% reported that they blogged a few times a week, 4.12% reported they blogged every day, and 17.53% had missing data.

The purported path model fit adequately to the data, χ^2^(12) = 20.12, *p* = 0.06, CFI = 0.94, TLI = 0.89, and SRMR = 0.07 (see Figure 1). Specifically, frequency of blogging at T1 was associated with T1 perceived blogging-related benefits, needs, and traits (β = 0.35, *p* < 0.001). T1 perceived blogging-related benefits, needs, and traits positively predicted T2 social support (β = 0.24, *p* = 0.03) and negatively predicted T2 memory slips (β = −0.23, *p* = 0.04). T2 social support positively predicted T3 mental health marginally (β = 0.14, *p* = 0.06). T2 memory slips negatively predicted T3 mental health (β = −0.15, *p* = 0.04). Autoregressive prediction of mental health from T1 to T3 was significant (β = 0.47, *p* < 0.001). Gender and age were not significantly associated with T3 mental health, *p*s > 0.05 (see Table 2).

The mediation processes were tested via bootstrapping based on 5000 bootstrap samples with replacement. The 95% confidence interval (CI) indicated that the standardized indirect effect between T1 frequency of blogging and mental health via T1 perceived blogging-related benefits, needs, and traits [CI: (−0.08, 0.09)], T2 social support [CI: (−0.02, 0.05)], and T2 memory slips [CI: (−0.11, 0.01)] included zeros. Hence, these variables did not mediate the relation between frequency of blogging and mental health. Next, the standardized indirect effect between T1 perceived blogging-related benefits, needs, and traits and T3 mental health via T2 social support [CI: (−0.004, 0.12)], and T2 memory slips [CI: (−0.007, 0.14)] included zeros. Hence, these variables did not mediate the relation between perceived blogging-related benefits, needs, and traits and mental health. Finally, the standardized indirect effect between T1 frequency of blogging and T2 social support via T1 benefits of blogging did not include a zero [CI: (0.002, 0.21)]. The 95% CI also indicated that the standardized indirect effect between T1 frequency of blogging and T2 memory slips via T1 perceived blogging-related benefits, needs, and traits did not include a zero [CI: (−0.19, −0.004)]. Hence, T1 perceived blogging-related benefits, needs, and traits mediated the relation between T1 frequency of blogging and T2 social support and memory slips, respectively.

Supplementary analysis with reversed directionality of effects between perceived blogging-related benefits, needs, and traits and blogging frequency indicated poor fit to the data, χ^2^(12) = 45.14, *p* < 0.001, CFI = 0.68, TLI = 0.42, and SRMR = 0.13. A further examination of the parameter estimates indicated that T1 perceived blogging-related benefits, needs, and traits were related to T1 frequency of blogging (β = 0.43, *p* < 0.001). However, T1 blogging frequency did not predict T2 social support (β = −0.03, *p* = 0.82) and T2 memory slips (β =0.04, *p* = 0.78). T2 social support did not predict T3 mental health (β = 0.03, *p* = 0.72). T2 memory slips negatively predicted T3 mental health (β = −0.27, *p* < 0.001). Autoregressive prediction of mental health from T1 to T3 was significant (β = 0.53, *p* < 0.001). Gender and age were not significantly associated with T3 mental health, *p*s > 0.05.

## 4. Discussion

Building on Miura and Yamashita’s theoretical tenets on blogging [12], this prospective study reveals a chain of processes between blogging frequency mental health. The findings also extended previous research on the benefits of blogging among undergraduate students in Hong Kong [15,16,23]. Of note, baseline frequency of blogging was associated with greater blogging-related benefits, needs, and traits. Blogging-related benefits, needs, and traits were further related to better social support and fewer memory slips three months later. Moreover, better social support marginally predicted better mental health, whereas fewer memory slips predicted better mental health three months later, after controlling for baseline mental health, age, and gender. Mediation findings further suggested that blogging-related benefits, needs, and traits served as a process between blogging frequency and social support, and between blogging frequency and memory slips. The findings advanced the literature by establishing the longitudinal relations between blogging frequency and social, cognitive, and psychological functioning during emerging adulthood.

The present study established a significant relation between blogging and its advantages, namely expression of thoughts and feelings and reflections of life experiences [9,12], both of which are crucial in emerging adulthood [31]. These findings also corroborated a recent study involving emerging adults in Hong Kong, which indicated the benefits of expressive writing on increased cognitive reappraisal [2]. The blogging-related benefits, needs, and traits were further associated with greater perceived social support. As emerging adults disclose themselves through blogging, they experience greater social support, potentially via the opportunities to interact with people who share similar interests or experiences [19,20]. Consistent with the literature [24], a marginal, though nonsignificant, trend further suggested that social support was associated with better mental health. Taken together, the findings corroborated previous research conducted in Hong Kong for the potential relevance of social support in blogging among emerging adults [23].

Aside from social support, bloggers were able to utilize their blogs as memory aids that allowed them to avoid experiencing reality disorientations [26]. By recollecting specific experiences and recognizing its benefits, blogging frequency was associated with fewer memory slips through the greater blogging-related benefits, needs, and traits. Consistent with the literature [27,28,39], those who have fewer memory slips had better mental health, such as fewer anxiety and depressive symptoms and a greater sense of self-assurance. In addition, given that the studies on memory slips primarily involved older adults [27,28,29] and clinical samples adults with PTSD [26] from the Western context, the present study adds to the literature by revealing the links between blogging, memory functions, and mental health among emerging adults from Hong Kong. 

Supplementary analyses further indicated that although blogging-related benefits, needs, and traits were related to greater social support and fewer memory slips, blogging frequency itself was not predictive of social support and memory slips. As such, the act of blogging may not be useful in enhancing individuals’ functioning. However, having blogged and recognizing its benefits, individual needs, and traits were crucial for better social and cognitive functioning. 

### Limitations and Future Directions

The present findings should be interpreted in light of several limitations. First, the study only included self-report. To minimize self-report bias, future studies should recruit multiple informants and use multiple methods, such as observations and physiological measures, to assess the variables of interest. Second, given that most of the participants were female, generalizability of the present findings is limited. Future studies should include a gender-balanced sample to increase generalizability. Relatedly, a larger sample may be necessary to increase the power to detect significant findings. Third, despite the longitudinal nature of this study, we did not have data for all of the variables across time points. Even though we controlled for T1 mental health and demographic covariates, future studies with a larger sample should collect data for all variables over time to reduce biases and assess measurement invariance. Fourth, unlike Miura and Yamashita’s model [12], in this study the blogging-related benefits, needs, and traits were entered to the model as a summative variable. In addition, the one-item measure of frequency of blogging was created just for this study. To understand the longitudinal mechanisms associated with blogging, future studies should increase the sample size and use well-validated measures. Fifth, in addition to 17.53% of missing values in the frequency of blogging, 34.02% of our participants were not bloggers. As such, a maximum of 51.55% of the participants might not have experienced any benefits of blogging. Future studies should include a larger sample of bloggers to gauge the precise benefits of blogging. Sixth, the present measure of mental health, i.e., Mental Health Continuum Short Form (MHC-SF) [35] assessed people’s emotional, social, and psychological well-being. Nevertheless, the measure did not capture psychological distress. Future studies should utilize measures of both mental health and psychological distress to more fully understand the role of blogging on the well-being among emerging adults. Seventh, in this study we did not examine the reversed directionality of effects from social support, memory functions, and mental health to blogging. Notably, the correlation between perceived blogging-related benefits, needs, and traits and mental well-being might be coincidental. When the non-bloggers had been removed from the data, the correlations between T1 blogging-related benefits, needs, and traits and T1 and T3 mental well-being remained significant (*r* = 0.56, *p* < 0.001 and *r* = 0.37, *p* = 0.001, respectively). Hence, even though some of the participants did not engage in blogging, they perceived its benefits, needs, and traits, which were further linked to better mental health. To disentangle the correlational effects between blogging, its perceived benefits, and individual functioning, large-scale longitudinal studies are needed. To further investigate the causal effects between blogging and individual functioning, experimental studies are needed. Finally, in this study we did not examine third variables such as family relationship quality, emotion regulation, and stress, which have previously been found to be correlated with emerging adults’ mental health [2,40,41]. To further understand the relation between blogging and individual functioning, future studies should take account of existing correlates of mental health. 

## 5. Conclusions

Supporting previous research in blogging particularly in an emerging adult context of Hong Kong [15,16,23], the present study calls attention to the longitudinal associations between blogging and psychological, social, and cognitive functioning among emerging adults. Notably, blogging frequency was linked to greater blogging-related benefits, needs, and traits, which w ere further linked to social support, memory slips, and subsequent mental health. Therefore, a chain of processes was identified between blogging frequency and mental health among emerging adults residing in Hong Kong. Interventions using blogs as a form of self-disclosure in promoting social support, memory functions, and mental health merit future investigation.

## Figures and Tables

**Figure 1 ijerph-20-05493-f001:**
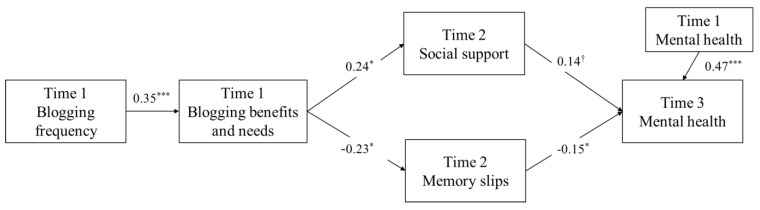
Final model between blogging and mental health, with gender and age as covariates of mental health. χ^2^(12) = 20.12, *p* = 0.06, CFI = 0.94, TLI = 0.89; SRMR = 0.07. Non-significant paths are not depicted in the figure for clarity. ^†^
*p* = 0.056, * *p* < 0.05, ^***^
*p* < 0.001.

**Table 1 ijerph-20-05493-t001:** Zero order correlations, means and SDs of the variables under study (N = 194).

Variable	(1)	(2)	(3)	(4)	(5)	(6)	(7)	(8)
(1) Gender (0 = male; 1 = female)	-							
(2) Age	−0.01	-						
(3) T1 Blogging frequency	0.16 *	−0.12	-					
(4) T1 Blogging-related benefits, needs, and traits	0.14	−0.20 *	0.35 ***	-				
(5) T2 Social support	0.10	−0.11	0.12	0.23 *	-			
(6) T2 Memory slips	0.02	−0.01	0.06	−0.18 *	−0.16 *	-		
(7) T1 Mental health	0.10	−0.10	0.04	0.43 ***	0.29 ***	−0.37 ***	-	
(8) T3 Mental health	0.20 *	−0.09	0.16	0.35 ***	0.32 ***	−0.33 *	0.60 ***	-
*M*	-	21.08	1.23	3.84	4.15	2.37	4.48	4.15
*SD*	-	2.00	1.28	0.67	0.74	0.59	0.87	0.92

Note. * *p* < 0.05, *** *p* < 0.001.

**Table 2 ijerph-20-05493-t002:** Standardized parameter estimates, unstandardized parameter estimates, and standard errors of the model.

Parameter	Standardized β	Unstandardized B (SE)	*p*
T1 Frequency of blogging			
→ T1 Blogging-related benefits, needs, and traits	0.35	0.17 (0.04)	<0.001
→ T2 Social support	0.04	0.02 (0.05)	0.65
→ T2 Memory slips	0.15	0.07 (0.04)	0.07
→ T3 Mental health	0.14	0.10 (0.06)	0.09
T1 Blogging-related benefits, needs, and traits			
→ T2 Social support	0.24	0.29 (0.14)	0.04
→ T2 Memory slips	−0.23	−0.23 (0.11)	0.04
→ T3 Mental health	−0.01	−0.02 (0.18)	0.93
T2 Social support			
→ T3 Mental health	0.14	0.17 (0.09)	0.06
T2 Memory slips			
→ T3 Mental health	−0.15	−0.23 (0.12)	0.04
T1 Mental health			
→ T3 Mental health	0.47	0.50 (0.10)	<0.001
Gender (0 = male; 1 = female)			
→ T3 Mental health	0.12	0.38 (0.21)	0.07
Age			
→ T3 Mental health	−0.00	−0.00 (0.03)	0.97

## Data Availability

The data presented in this study are available on request from the corresponding author. The data are not publicly available due to ethical constraints.

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
