# Peer review of "Do Bloggers Have Better Mental Health? The Social, Cognitive, and Psychological Benefits of Blogging in Emerging Adulthood"

_ijerph, 2023, doi:10.3390/ijerph20085493_

Round 1
Reviewer 1 Report
I read this paper with great interest and have several suggestions to the authors to address.
First, I would like to see more text on the context and the topic in HK and EAs in this context.
Second, what is the incremental contribution and relevance and novelty for EA field in general and in HK?
Third, any missing values analyses and longitudinal invariance?
Author Response
Please refer to the attached responses to Reviewer 1's comments. Thank you.

Reviewer 2 Report
The manuscript describes a study on the relationship between blogging and mental health. This topic is interesting for the readers of the IJERPH.
The Introduction gives a good and concise overview on the relevance of the topic. While it is to be expected that social support is related to blogging, the role of memory slips is less clear, and the literature references also cannot clarify the meaning. The reference to intensive care units is a little bit far from the situation of young bloggers.
Methods: As in most online surveys, it is difficult to assess the generalizability of the study population. The number of 194 is not very high, and when using more sophisticated statistics such a s path analyses, the limitations of this sample size become noticeable. However, for the analysis of rough relationships the sample size may be sufficient.
line 93: n(T2)= 180, n(T1)=194, this results in a response rate of 180/194=92.78 % and not 91.83 %. By chance it is the same response rate as that from T2 to t3.
Measures:
The assessment instruments for the main topics (Frequency of blogging and Benefits and Needs for Blogging) are not well-established instruments, but probably there are no better elaborated instruments available, and the questions of the first two scales and the three additional questions are largely reasonable. However, I see the problem that questions on self-consciousness and reassurance seeking are not related to blogging: “I reflect about myself a lot” is not a question on benefits of blogging.
I assume that each of the four scales of the Benefits instruments consist of two items, if so, it should be written down.
The instrument for measuring mental health seems not to be optimal since it measures kinds of well-being, which is not completely identical with mental health. In my understanding mental health should mainly include the absence of mental problems (low levels of anxiety and depression), but this point is not problematic.
Results
The mean score of the frequency of blogging is 1.23, measured with a 1-5 scale. This means that at least 77 % of the participants responded with 1 (=never). This is only the case if all other participants responded with “2”; if there were more intensive bloggers, the number of never-bloggers would even increase.
This is my main concern: If the sample comprises mostly non-bloggers, what is the meaning of their responses to “Blogging benefits and needs”? These non-bloggers may guess what benefits blogging might have if one would perform blogging, but the authors treat the variable as reflecting real experiences.
The mean of “Blogging benefits and needs” is also difficult to understand. The authors write that the scores are averaged, but then the scale range should be 1-5, and a score of 29.84 is impossible.
Table 1 presents zero order correlations between the variables, and I very much appreciate the indication of these correlations (and not only the presentation of the results of the complex path analysis). In Table 1 we find a statistically significant negative correlation between frequency of blogging and blogging benefits (r = -0.23). This is not very plausible: Those who are (voluntarily) blogging see more problems than benefits in blogging. Perhaps this correlation is due to the fact that the sample mainly consists of non-bloggers.
And then, in the path model, we see a highly significant positive association (beta = 0.35).
It is possible that the beta coefficients have been calculated correctly within the path model, but this discrepancy between a significant positive and a significant negative association between the same pair of variables indicates a certain instability, and the difficulty to interpret the results.
Figure 1: I would prefer to put the box with Time1 Mental health to the left side of the figure so that the figure consequently illustrates the temporal course.
Table 2: One beta=.14 is indicated as (nearly) statistically significant (p=0.06), and the other beta=.14 (T1 frequency blogging => T3 mental health) has no indication of statistically significance. This seems to be inconsistent: What is the significance of this beta=0.14?
line 180 “between frequency of blogging and mental health via..” please add the time of the mental health measurement, e.g. “… and mental health at T3 via…”.
line 196: I do not agree that frequency of blogging is associated with greater benefits. Table 1 shows the opposite: frequency of blogging is associated with lower benefits. Only if one uses a sophisticated system with several assumptions on the underlying path the relationship becomes positive.
Discussion of the relationship between blogging benefits, low frequency of memory slips, and mental health: The study proved that these three variables are interrelated. However, this relationship may also be due to associations between these three variables and a further underlying variable such as a tendency to respond to questionnaires in a positive way. People with a positive tendency see benefits in blogging, report good own memory and good mental health. The path model does not prove the causal relationships. However, this is not stated in the discussion, the discussion is restricted to associations, which is correct. Nevertheless, while I understand the relevance of the relationship between blogging and social support, I do not see the relevance of the memory quality in the context of blogging.
Limitations: Several limitations are given in this section, however, this review presents some further limitations.
References
Sometimes there is a “,” and sometimes a “;” between the author names.
Sometimes journal names are abbreviated and sometimes not.
Author Response
Please refer to the attached responses to Reviewer 2's comments. Thank you.

Reviewer 3 Report
The reviewed paper aimed to “investigate the longitudinal processes by which blogging-related disclosure is linked to mental health”.
The article uses relevant literature on the subject. A proper selection of sources was made. However, in the review of the subject literature conducted, the most recent publications (e.g., 2018, 2019, 2020, 2021, 2022) should be taken into account to a greater extent.
The research course of action and the methods and techniques of analysis adopted are adequate for this type of research. The analysis of the results of the research was conducted properly. The correct interpretation of the obtained research results was made.
In the conclusion, it is worth clearly indicating whether the stated purpose of the work was realized.
In addition, I propose checking certain aspects of bibliographic reference lists. For example, the journal’s name should be written with its abbreviation, the volume in italics, and the year in bold, and the text should be justified. There are several shortcomings. It needs improvement.
Despite the indicated limitations, the scientific value of the article can be assessed positively.
Author Response
Please refer to the attached responses to Reviewer 3's comments. Thank you.

Round 2
Reviewer 2 Report
The authors have adequately addressed many of the points raised. However, shortcomings remain.
The authors now report the frequencies of responses to the item Frequency of blogging. From these frequencies, it appears that 66 people never blog, and that there is a miss there for another 34 people.
First of all, if the frequencies given are correct, then the mean value given in Table 1 is wrong. It would then not be 1.23, but 2.23. It is disappointing that the authors, although they were pointed to the conclusion from the mean value of 1.23 in my previous review, did not correct this.
Furthermore, missings values in the variable of frequency of blogging means that these individuals cannot be included in the correlation and regression analyses, so these analyses are based on far fewer than the 194 individuals reported.
Since the authors now report that there are many missing values in the variable of frequency of blogging, it is natural to ask how many missing values there are in the other blog-related variables. A statement on this would help to better assess the credibility of the results.
Minor comments:
There are now significance statements in Table 2. The statements p<0.000 are meaningless, because a probability can never be smaller than zero.
The text says n=194 (which is not quite correct given the relevant missing values), but it should be compatible with the abstract, and there it says n=192.
In the limitations it says that 34.02% were non-bloggers. This suggests the expectation that the rest, about 66%, were bloggers, but that's not true because of the missing values. It would be fair to add here in the limitations that in addition to non-bloggers, there were also 17.53% people with missing values in the frequency of blogging.
Through revision, we now learn that the sample size of those who blog at all is only 94, less than half of the stated 194 people. From my point of view, as already described in the first review, this is too few for a publication. However, if the editor and the other reviewers are in favor of a publication, then at least the remaining errors should be cleared up.
